# Research on the Comprehensive Regulation Method of Combined Sewer Overflow Based on Synchronous Monitoring—A Case Study

Lei Yu [1,2,*], Yulin Yan [1,2], Xingyao Pan [1,2], Simin Yang [1,2], Jiaming Liu [1], Moyuan Yang [1,2] and Qingyi Meng [1,2]

1 Beijing Water Science and Technology Institute, Beijing 100048, China
2 Beijing Unconventional Water Resources and Water Saving Engineering Technology Research Center, Beijing 100048, China
* Correspondence: yl@bwsti.com

**Abstract:** Combined sewer overflow pollution has gradually become the limiting factor for the further improvement of river water quality during rain events. Setting up a comprehensive regulation method based on synchronous monitoring is essential for combined sewer overflow management. However, current studies mainly focus on single monitoring and lack a correlation between control objectives and control effects. This study establishes a new aspect of a comprehensive regulation and control method based on overflow characteristic analysis, a calculation model, and control target determination. Through synchronous monitoring of the pipe network, the sewage treatment plant, and the river course in the Liangshui River basin of China, rainfall thresholds of outlets in a combined pipe network, pre-treatment overflow, and simple-treatment overflow were 14, 9, and 16 mm, respectively, and the overflow volume was positively correlated with the rainfall. The COD (chemical oxygen demand) concentration from the pre-treatment overflow was much higher than that from the combined pipe network, and the EMC (event mean concentration) in heavy rain was higher than in rainstorms. The shortest time exceeding the water quality by overflow pollution was 1 h, and the longest time was more than 7 days. Overflow load proportions of the three links were 43.4%, 32.8%, and 23.8%, accounting for 66.3% of the total pollutant load of the river, and the best scheme of input–output ratio was to regulate the first three outlets of overflow load. Our results provide comprehensive guidance and a systematic approach for the monitoring and control of combined sewer overflow.

**Keywords:** combined sewer overflow; synchronous monitoring; Liangshui River; rainfall characteristics; comprehensive regulation

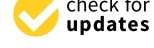



## 1. Introduction

Combined sewer overflow (CSO) refers to the phenomenon where surface runoff generated by rainfall flows into the combined sewer system; when the flow exceeds the interception, the mixed sewage will be directly discharged into the receiving water body, thus polluting the water quality. According to the China Urban Construction Statistical Yearbook in 2019 [1], the total length of drainage pipelines in China is 743,981.9 km, of which combined pipelines are 103,776.17 km, accounting for 13.95%. The highest proportion of combined pipelines in each region is 60.52%, and the lowest is 5.01%. The length of combined pipelines in Beijing is 1528 km, accounting for 8.49%, mainly distributed in the functional core areas of the capital (Dongcheng District and Xicheng District). With the promotion of two three-year pollution control schemes in Beijing, point source pollution has been effectively controlled, and CSO pollution has become the main factor restricting the further improvement of river water quality in the urban area of Beijing [2,3]. In 2021, the People's Government of Beijing Municipality issued the Implementation Plan for the Prevention and Control of Urban Waterlogging and Overflow Pollution in Beijing

(2021–2025) (Beijing Administration Office No.6, 2021), which determines the objectives and indicators of CSO control; the sewage will not flow into the river when the rainfall is less than 33 mm at the overflow and crossing outlets in the central urban area by 2025.

CSO-related research in foreign countries has already started [4–6], and CSO has gradually become a hot topic of research in China due to the development of black-odorous water treatment and sponge city construction. Jia Nan et al. [7] and Li Junqi et al. [8] reviewed the control standards and indicators of CSO in Europe and the United States. In the long-term control plan of CSO in the United States, the annual overflow frequency (control standard is 4–6 times), overflow control ratio (control standard is > 85%), and pollutant removal rate (control standard is TSS (total suspended substance) > 50%) are the main control indicators. European countries have adopted different indicators and standards; the Netherlands and Belgium use overflow frequency as control indicators, with control standards being 3–10 and 7 times, respectively. In Britain, bacterial contamination, the ammonia standard, and the oxygen standard are used as control indicators. At present in China, control indicators mainly consist of rainfall control (33 mm in Beijing), overflow frequency (10 times in Huangxiao River in Wuhan) [9], and the overflow pollution control rate (80% in Shanghai, corresponding to rainfall of 19 mm). Li He, Li Siyuan et al. [10–13] have studied the CSO pollution characteristics in Shanghai, southern Jiangsu, Zhuhai, the upper reaches of the North Canal in Beijing, and other regions, and the results have shown that it is difficult to monitor and identify CSO characteristics due to high flows, short durations, and high pollutant content [14]. The overflow characteristics are affected by many factors, including rainfall, underlying surface, and pipe network interception. The sewage concentration changes greatly in the process of overflow, which is closely related to rainfall and regions. Yan Pan et al. [15], He Junchao et al. [16], and Wang Haozheng et al. [17] discussed the strategy of CSO control and the connection and characteristics of drainage systems in China, demonstrating that CSO treatments mainly include source reduction, pipe network diversion, river interception, mobilization and storage before estuaries, and expansions of sewage plants.

To sum up, two main deficiencies exist in the current research on CSO. Firstly, in the aspect of CSO monitoring, the research focuses primarily on outlets and lacks synchronous analysis in rivers and sewage plants. Water sampling of outlets mostly depends on manual work, where accuracy and frequency are hard to guarantee; flow quality and quantity create difficulties for synchronous monitoring. The frequency of river quality is mostly monitored by days; thus, the regularity of river quality during rainfall cannot be accurately reflected. The above problems restrict the identification of CSO regularities and weaken the association between CSO pollution and river quality. In the other aspect of CSO control, the research focuses mostly on theoretical and technical paths in which CSO reduction is based on models to quantify the contribution of source, process, and end facilities [18–20]. Moreover, CSO project cases mainly concentrate on the construction of storage tanks [21–23] and lack universal methods and comprehensive regulation of the outlet, pipe network, sewage plant, and river.

In search of a new approach to solve deficiencies in the current research of CSO and to formulate an appropriate control scheme based on the reasonable analysis of pollutant loads in different links, this study explores the regularity of overflow occurrence in the main link of combined sewer systems and establishes an overflow model based on CSO coefficients via the synchronous monitoring of water quality and quantity in pipe networks, sewage plants, and rivers. The relationship between the scale and over-standard duration of river quality is established by analyzing the water quality load in the over-standard period of the river. To support the selection of control schemes under different management regimes, a set of comprehensive regulation methods of CSO is formed and applied to the central section of Liangshui River in Beijing based on the synchronous monitoring of sewage treatment plants, networks, and rivers. The results can support CSO governance in Beijing and also provide a reference to other cities in China.

## 2. Materials and Methods

### 2.1. Overview of the Study Area

Liangshui River is one of the four major drainage rivers in the central city of Beijing in China; it is located in the south of the central city. It originates from Shougang Retreat Canal in Shijingshan District and enters the North Canal at Yulin Village in Tongzhou District. The total length of the main stream is 68.41 km, and the main tributaries include Shuiya River, Xinfengcao River, Han River, and Macao River, with a drainage area of 629.7 km². It is the largest of the four major basins in the central city of Beijing, with a climate of temperate monsoon. The average rainfall in the past ten years (2008–2017) has been 545 mm, which is mainly concentrated in the flood season, from June to September. The urban section of Liangshui River is selected as the study area, with an area of approx. 189 km² (as shown in Figure 1). The terrain of the region gradually decreases from northwest to southeast, and the underlying surface is rich in types, including old urban areas, newly built areas, and suburban areas. In recent years, through sewage interception, river dredging, ecological protection, and other projects, water quality in the urban section of Liangshui River has attained the required standards [24].

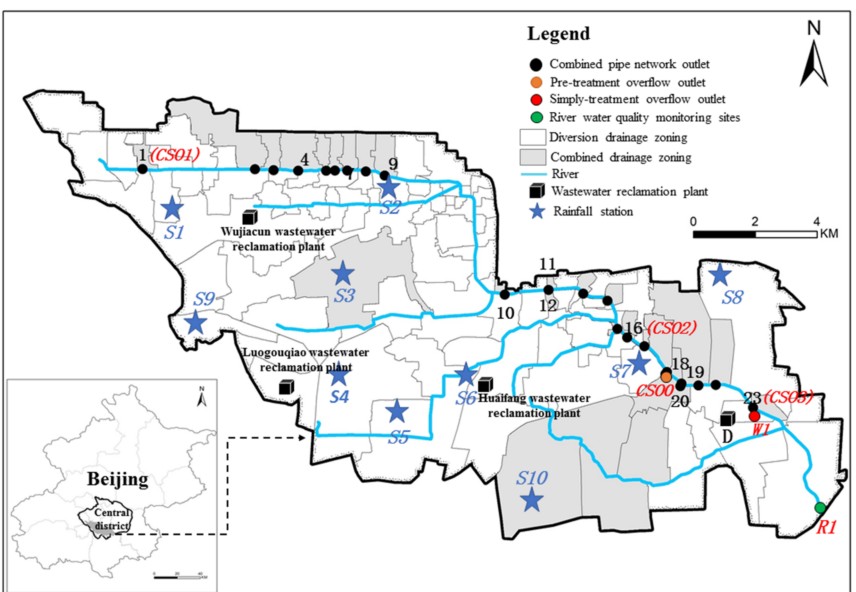

**Figure 1.** Location and monitoring points of Liangshui River basin.

The mixed sewage system in the study area includes a separate sewer system and a combined sewer system, and the area of the combined sewage system accounts for about 1/4 of the total. According to the survey, 23 combined drainage outlets, with circular and rectangular sections, are located on both sides of the main stream of Liangshui River (Figure 1, numbered 1–23 from top to bottom). The size of the outlets ranges from 800 to 4000 mm. Weir flow interception is adopted for all outlets, with an interception ratio of 1. The total capacity of the four wastewater plants (shown in Figure 1) of the basin is 1.38 million cubic meters per day; they can be interconnected through the main sewage pipes to realize the allocation of each wastewater plant. Due to the limitations of the treatment process, only terminal sewage plant D can cope with the impact of overloaded sewage in a short time. When rainfall occurs and the upstream sewage plant is fully loaded, the sewage in the region will enter sewage plant D through dispatching. After exceeding the treatment load of sewage plant D, it crosses and discharges into Liangshui River. An overflow outlet in front of the plant is 3 km away from sewage treatment plant D. Overflow will occur when the inflow exceeds the threshold to ensure the safety of sewage treatment plant D.

CSOs in the study area mainly occur at three link points (as shown in Figure 2): (1) outlet overflow, which occurs at the end of the intercepting well when pipeline capacity

is exceeded; (2) pre-treatment overflow: the combined sewage intercepted to the pipe exceeds the transmission capacity or the water level of the downstream sewage treatment plant is too high; (3) crossing discharge: sewage exceeding the capacity of the sewage plant is directly discharged into the river after passing the coarse bar screen (only floatable removed). All three links are referred to as overflow discharge.

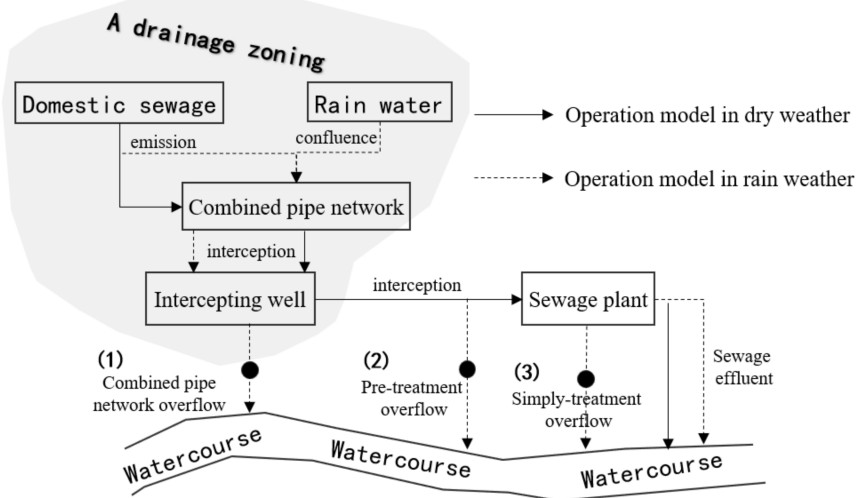

**Figure 2.** Overview of CSO drainage system in the study area.

### 2.2. CSO Coefficient-Based Comprehensive Regulation and Control Technology for Combined Overflow System

#### 2.2.1. Technical Process

Through data collection and field surveys, information on the rainfall, underlying surface, and catchment areas in the study was obtained. The representative overflow link of the combined sewer system was selected to monitor the synchronous water quantity and quality of PSR (pipe network, sewage treatment plant, and river course). Based on the monitoring data, we established the overflow regularity, a calculation model (see Section 2.2.2 for details), and the load equation, which calculates the volume and load of each overflow link. Then, the time of water quality exceeding the standard was analyzed in controlled rainfall events, calculating the hourly proportion of pollutant load. Different control schemes could be formulated according to the overflow load of each link. On the basis of the same proportional reduction, water quality in different storage volumes and time exceeding the standard was calculated; then, we established the relationship between the time and scale of regulation. Finally, the regulation scheme was determined based on the relationship and requirements. The technical flow chart is shown in Figure 3.

#### 2.2.2. Calculation Model of Overflow

With the different links and fixed catchments in the combined sewer system, the overflow discharge is mainly affected by rainfall, so the overflow under different control rainfall situations can be obtained through the correlation. As the overflow of outlets is affected by rainfall and catchment areas, we are unable to calculate the overflow of each outlet under different control targets by the correlation between rainfall and overflow. For this reason, the concept of a "CSO coefficient" is put forward, which is the ratio of overflow depth R to precipitation depth P in the combined catchment in any period (Equation (1)). Based on the "CSO coefficient", an overflow calculation model of outlets (Equations (2) and (3)) is established. The detailed process is as follows:

(1) Based on the monitoring data, the CSO coefficient $\psi_{cso_{(m,i)}}$ was obtained under different rainfall situations in the catchment area of each outlet according to Equation (1). (2) By carrying out linear fitting on the rainfall and the CSO coefficient through the least square method, the relationship shown in Equation (2) was established. (3) The corre-

sponding CSO coefficient was obtained according to the control target (namely, the control rainfall). (4) The overflow of different combined drainage zones can be calculated according to Equation (3).

$$\psi_{cso_{(m,i)}} = \frac{Q_{cso_{(m,i)}}}{10R_{(m,i)}F_m} \tag{1}$$

$$\psi_{cso} = a + bR \tag{2}$$

$$Q_{cso_m} = 10 \times R \times F_m \times \psi_{cso} \tag{3}$$

where $\psi_{cso}$ is the CSO overflow coefficient, dimensionless; $Q_{cso}$ is the overflow, m$^3$; $R$ is the rainfall per event, mm; $F$ is the area of the catchment corresponding to the combined system outlet, hm$^2$; $m$ is the outlet number; $i$ is the field of the number of rainfall events.

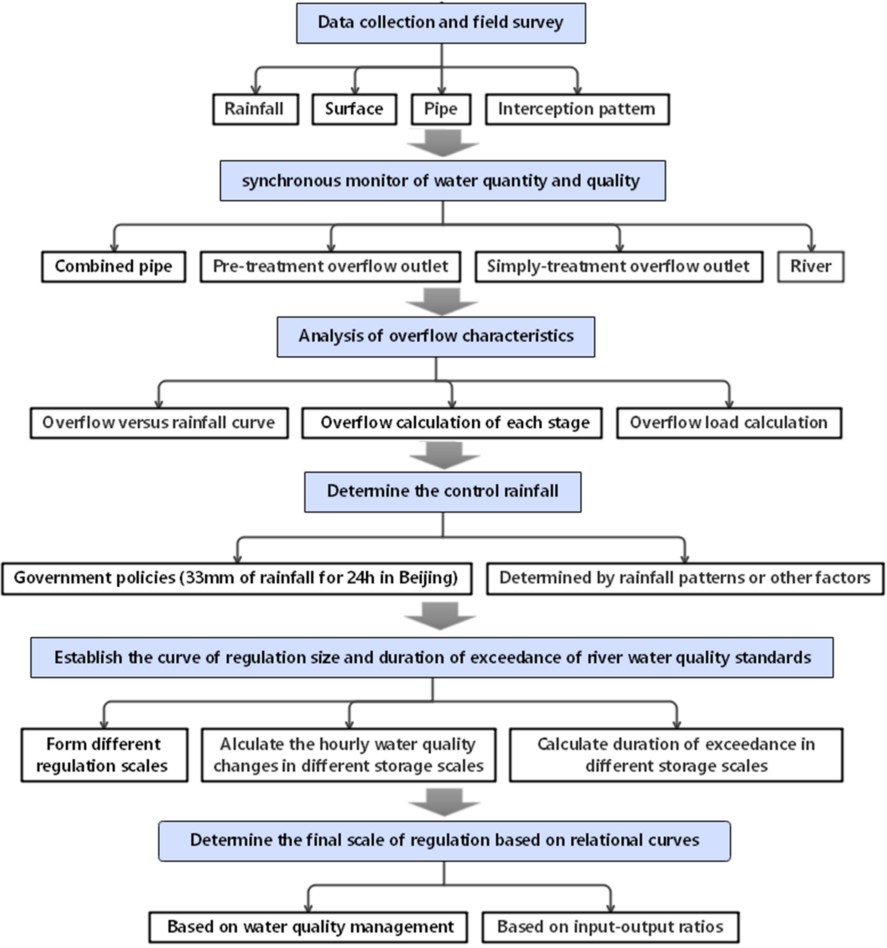

**Figure 3.** Technical process of the CSO comprehensive regulation method based on the synchronous monitoring of PSR.

2.2.3. Calculation Method of Overflow Load

The overflow load is the product of the overflow volume and concentration in Equation (4). Considering that the concentration of overflow varies greatly in the process of rainfall, the event mean concentration (EMC) of rainfall pollutants is used to analyze the water quality at the outlet; the calculation equation of EMC is shown in Equation (5).

$$W_{cso_m} = Q_{cso_m} \times EMC_m \tag{4}$$

$$EMC = \frac{\sum_{i=1}^{n} C_i \cdot V_i}{\sum_{i=1}^{n} V_i} \tag{5}$$

where $W_{cso_m}$ is the overflow load at *m* outlet, kg; $C_i$ is the concentration of the pollutant sampled within *i* sections, mg/L; $V_i$ is the runoff during the sampling period, L; *n* is the sampling number of the entire rainfall.

### 2.3. Monitoring Plan

Based on the upstream drainage of outlets and types of underlying surface (Table 1), three outlets in the combined sewer system were selected for monitoring (CSO1, COS2, and CSO3 in Figure 1). The pre-treatment overflow outlet (CSO0) and sewage treatment plant crossing (W1) were monitored at the same time. One water quality monitoring point, numbered R1, was set at the end of the river. Ten rainfall monitoring stations were set up to reflect the spatial distribution of rainfall in the whole study area, which were numbered S1 to S10, and the rainfall data were monitored every 5 min.

**Table 1.** Information of catchment area corresponding to each monitoring point.

| Outlet | Area (ha) | Area of Each Type of Underlying Surface (ha) | | | | |
| --- | --- | --- | --- | --- | --- | --- |
| | | Greenbelt | Bare Land | Water Area | Architecture | Road |
| CSO1 | 238.0 | 38.8 (16.3%) | 7.1 (3.0%) | 0 | 163.3 (68.6%) | 28.8 (12.1%) |
| CSO2 | 34.7 | 0 | 0 | 0 | 33.3 (96.0%) | 1.4 (4.0%) |
| CSO3 | 152.6 | 9.3 (6.1%) | 12.2 (8.0%) | 0 | 115.2 (75.5%) | 15.9 (10.4%) |
| CSO0 | 13,358.3 [1] | 1375.9 (10.3%) | 440.8 (3.3%) | 40.1 (0.3%) | 9858.4 (73.8%) | 1643.1 (12.3%) |
| W1 | 18,879.8 [1] | 2718.7 (14.4%) | 887.4 (4.7%) | 94.4 (0.5%) | 13,083.7 (69.3%) | 2095.7 (11.1%) |

[1] CSO0 and W1 are the water collection range corresponding to the pre-treatment overflow outlet and the sewage treatment plant crossing.

The weir-type flowmeter was selected to monitor the flow of outlets every 5 min, and a self-developed intelligent sampler (patent No.: ZL201720745546.9) was selected to monitor the water quality. When the overflow occurs from the outlet, the automatic sampler collects the effluent stored in a 500 mL polyethylene bottle. The sample collection interval was set as follows: samples were taken every 5 min in the first half an hour after overflow, and every 10 min from half an hour to 1 h, then every 30 min after 1 h of overflow, until all the 24 bottles in the sampler were full or the overflow was over.

An online flowmeter was used to monitor the flow of the simple-treatment overflow outlet every 1 h. The equipment, based on the principle of quantum dot spectral sensing technology [25], was used to monitor river quality every 10 min, and river flow was monitored via the nearby hydrological station every 1–2 h.

### 2.4. Acquisition of Data Summary

During the study period, there was a total of 13 effective rainfall events (rainfall greater than 0.2 mm accumulated in 24 h), of which 7 events obtained the monitoring data of overflow flow and water quality, 4 obtained the data of sewage treatment plant crossings, and 7 obtained the data of river quality and flow. All the monitoring data are summarized in Table 2.

**Table 2.** Summary of monitoring data.

| Serial No. | Type of Monitoring Data | Index | Quantity | Time | Resolution |
|---|---|---|---|---|---|
| 1 | Rainfall data | Rainfall Rainfall time | 10 sites 13 events | 2020.7–2020.9 | 5 min |
| 2 | Flow of pipe network outlet and pre-treatment overflow outlet | Flow rate | 4 outlets 7 events | 2020.7– 2020.9 | 5 min |
| 3 | Water quality of pipe network outlet and pre-treatment overflow outlet | COD | 4 outlets 7 events | 2020.7–2020.9 | 5 min (0–0.5 h) 10 min (0.5–1 h) 30 min (1–8.5 h) |
| 4 | Simple-treatment overflow volume | Flow rate | 4 events | 2020.7–2020.9 | 1 h |
| 5 | River quality and flow | COD Flow rate | 10 events | 2020.7–2020.9 | 5 min (water quality) 1–2 h (flow) |

## 3. Results and Discussion

### 3.1. Analysis of CSOs in Rainfall Events

To reflect the relationship between rainfall and overflow more accurately and to consider the spatial distribution of rainfall, the network overflow was analyzed by the nearest rainfall station in its catchment area. As the pre-treatment and simple-treatment overflows collect the drainage of the upstream basin, the average rainfall of 10 stations was analyzed. Hence, CSO1 adopted the rainfall data of S1, CSO2 and CSO3 adopted the data of S7, and CSO0 and W1 were related to the average rainfall.

As CSO is mostly affected by many factors, such as rainfall, catchment area, underlying surface, interception ratio, and the capacities of the sewage network and sewage treatment plants, in this study, the interception ratio, catchment area, and capacity of sewage were all determined, so rainfall was the common factor affecting the overflow of the three links. Moreover, the network overflow was also affected by the catchment area.

All the monitoring rainfall data related to the overflow are summarized and analyzed in Table 3, in which the average rainfall intensity is the ratio of one rainfall to the duration of rainfall. The rainfall was graded according to the Precipitation Grades (GB/T 28592-2012); the 13 rainfall events included 5 light rain events, 2 moderate rain events, 4 heavy rain events, 1 rainstorm, and 1 heavy rainstorm, covering all rainfall grades.

**Table 3.** Summary of rainfall characteristics and overflow situation.

| Rainfall Field No. | Rainfall Range (mm) | Average Rainfall Duration (h) | Mean Rainfall Volume(mm) | Average Rainfall Intensity (mm/h) | Overflow or Not | | |
|---|---|---|---|---|---|---|---|
| | | | | | Combined Pipe Network Outlet | Pre-Treatment Overflow Outlet | Simply-Treatment Overflow Outlet |
| 1 | 0–3.5 | 2.88 | 1.18 | 0.41 | No | No | No |
| 2 | 3.0–28.0 | 0.70 | 16.62 | 23.74 | No | Yes | Yes |
| 3 | 0.7–12.5 | 1.78 | 3.94 | 2.21 | No | No | No |
| 4 | 17.0–29.0 | 1.02 | 23.15 | 22.77 | Whole | Yes | Yes |
| 5 | 56.5–111.9 | 12.43 | 89.98 | 7.24 | Whole | Yes | Yes |
| 6 | 2.0–17.5 | 8.35 | 9.36 | 1.12 | Part | Yes | No |
| 7 | 23.5–71.0 | 10.23 | 52.23 | 5.10 | Whole | Yes | Yes |
| 8 | 0–10.0 | 4.35 | 3.69 | 0.85 | No | No | No |
| 9 | 3.0–24.0 | 19.82 | 10.96 | 0.55 | No | No | No |
| 10 | 0.5–5.5 | 0.75 | 2.46 | 3.28 | No | No | No |
| 11 | 0.5–5.5 | 1.70 | 1.41 | 0.83 | No | No | No |
| 12 | 12.8–37.0 | 6.13 | 22.65 | 3.69 | Whole | Yes | No |
| 13 | 12.5–46.5 | 8.78 | 23.06 | 2.63 | Whole | Yes | No |

By analyzing the overflow conditions under different rainfall levels, overflow would occur in rainstorms and levels above rainstorms at all three links in the combined system, while none of the links overflowed in light rain. In heavy rain, all outlets in the combined system and the pre-treatment area would overflow; the sewage treatment plant would overflow in 50% of rainfall events. Pre-treatment overflow occurred in moderate rain, where only part of the network would overflow. The relationship between rainfall and overflow is further analyzed and shown in Figure 4. The threshold value of rainfall overflow in CSO0 was the lowest, which was 9 mm; when the rainfall reached 14 mm, CSO1, -2, and -3 would overflow. The threshold value in W1 was greater than 16 mm. Hence, the threshold value of rainfall overflow in the pre-treatment overflow was the lowest among the three links, followed by the combined system and then the simple-treatment overflow; this is corroborated with the analysis of rainfall grades.

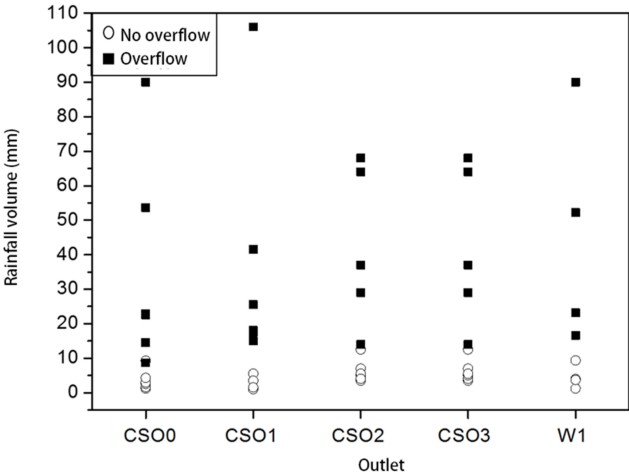

**Figure 4.** Relationship between rainfall and overflow at each monitoring point.

In conclusion, in moderate rain or above, the pre-treatment outlets and most of the network outlets would overflow; in heavy rain or above, overflow would occur in the sewage treatment plant. Therefore, when overflow occurred in the sewage treatment plant, the pre-treatment and combined network outlets would also overflow.

### 3.2. Analysis of Influencing Factors and the Calculation Model of Overflow

SPSS software was used to analyze the Pearson correlation among overflow, total rainfall, maximum five-minute rainfall, average rainfall intensity, and catchment area. The analysis (Table 4) showed that the overflow was significantly correlated with rainfall but not with maximum five-minute rainfall and average rainfall intensity. By comparing the correlation coefficients of CSO1, CSO2, and CSO3 with rainfall, the correlation decreased with an increase in catchment area. The overflow presented a significant correlation with the catchment area of each outlet, with a correlation coefficient of 0.96[2], indicating that in the same rainfall, the larger the area, the more the overflow.

**Table 4.** Correlation coefficient between overflow and rainfall.

| Outlet | Rainfall Volume (mm) | Maximum Rainfall in Five Minutes (mm) | Average Rainfall Intensity (mm/h) |
|--------|----------------------|----------------------------------------|-----------------------------------|
| CSO1 | 0.706 [1] | 0.606 [1] | 0.172 |
| CSO2 | 0.965 [2] | 0.339 | 0.267 |
| CSO3 | 0.895 [2] | 0.161 | 0.132 |
| CSO0 | 0.953 [2] | 0.321 | 0.334 |

[1]: Significant correlation at the 0.05 level (two-tailed); [2]: significant correlation at the 0.01 level (two-tailed).

CSO coefficients of each rainfall in the three combined sewer outlets were calculated based on the model of combined overflow, and the least square method was used to linearly fit the CSO coefficient and rainfall. These results, including the overflows of pre-treatment and simple treatment fitted with rainfall, are jointly shown in Figure 5 and emerge as excellent fitting results, with coefficients of more than 0.9.

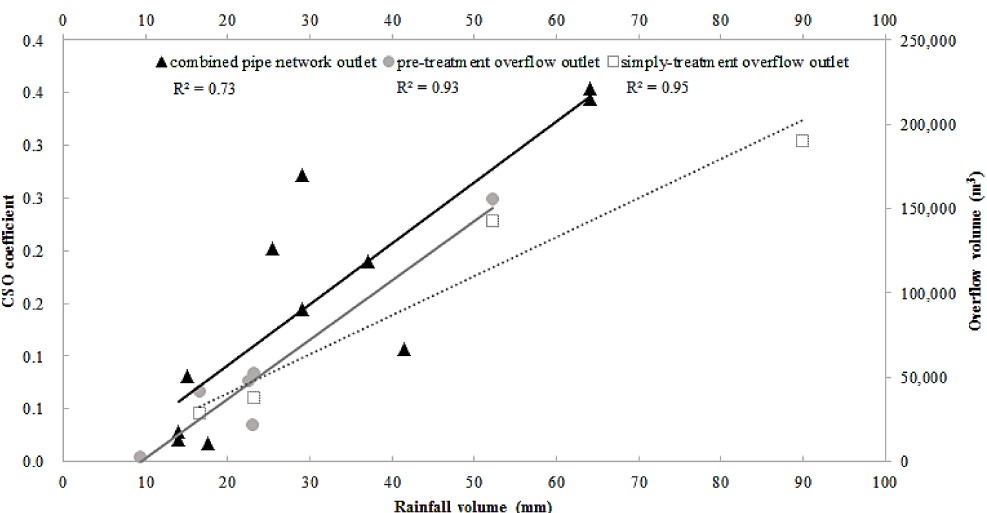

**Figure 5.** Fitting curve of rainfall with CSO coefficient, CSO0 overflow, and W1 overflow.

The EMC values calculated by the COD of each outlet in rainfall events 4, 7, and 12 are shown in Figure 6. The order of EMC values was CSO0 > CSO1 > CSO3 > CSO2, and the water quality in the pre-treatment outlet was much higher than in the combined network. The comparative analysis of rainfall events demonstrated that EMC values in heavy rain (rainfall events 4 and 12) were higher than in heavy rainstorm (rainfall event 7), indicating the rainfall grade (rainfall amount) could affect the water quality of overflow; this conclusion corresponds to a study on stormwater detention tanks that concluded it was impossible to effectively control combined sewer overflows by only treating the initial rainwater [26].

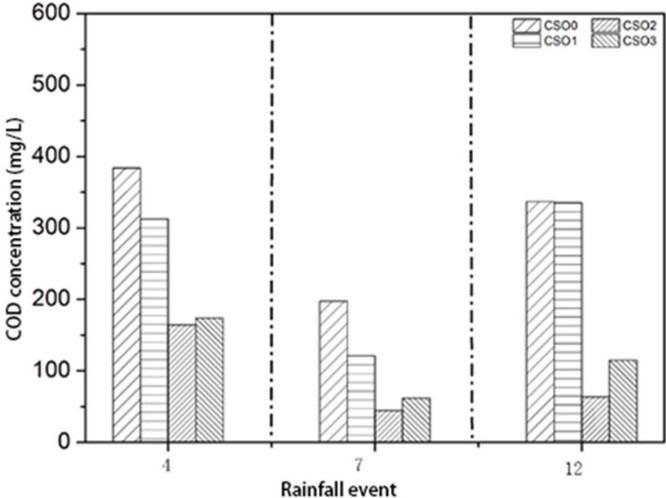

**Figure 6.** EMC of each monitoring point in a typical rainfall event.

When under the same rainfall grade, the EMC value in rainfall event 4 was greater than in rainfall event 7, in which the former rainfall intensity was 7.5 times that of the latter, presenting that the water quality could also be affected by the rainfall intensity.

### 3.3. Analysis of the Impact of CSOs on River Quality

Using the Class V standard of surface water (concentration of COD equal to 40 mg/L) as a comparison, the river quality was continuously monitored for two months, as shown in Figure 7. Rainfall events 4 and 12 were both heavy rains, but the former was 6 times the latter in rainfall intensity; by contrast, the maximum concentration and duration of rainfall event 4 were 2 and 3 times, respectively, those of rainfall event 12, proving that pollutant loads were affected by rainfall intensity. Hence, in the same rainfall condition, the greater the average rainfall intensity, the higher the overflow pollutant load, which is consistent with the conclusions in Section 3.2. The most serious exceedance of river quality happened in rainfall 7, but this pollutant load from the combined sewer outlets was lower than events 4 and 12, according to the EMC analysis in Section 3.2. In consequence, besides the overflow pollution from the combined sewer system, there were other sources of pollution in the river; this conclusion is consistent with related studies on pollution levels in stormwater during rain events [27].

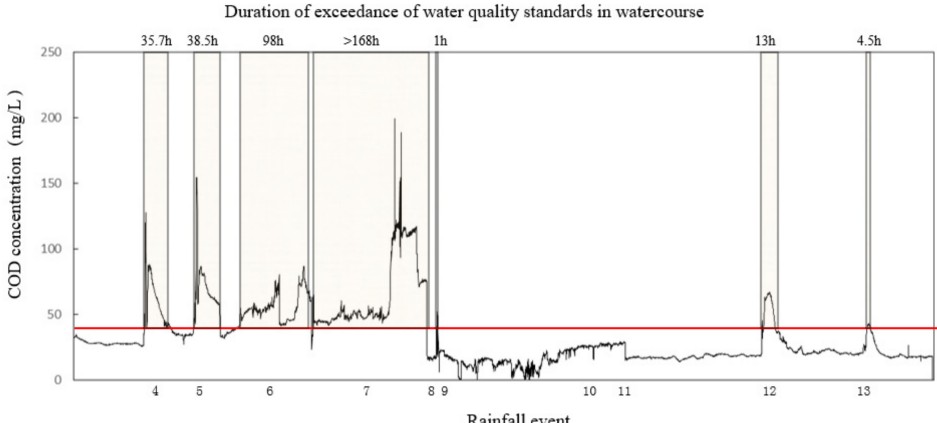

**Figure 7.** Impact of rainfall on river water quality (red line is the surface water Class V standard for COD, 40 mg/L).

There were four events and rainfall intervals of 72–110 h from rainfalls 4 to 7, in which the period that river quality reached the standard was only 47 h, and the longest exceedance was for more than 7 days. Due to a lack of time for self-purification in the river, water quality appeared to have a long time of exceedance. As the interval between rainfall events 9 and 10 was nearly 10 days, and there was less rainfall in rainfall events 10 and 11, without combined overflow, river quality up to the standard reached nearly 20 days from rainfall events 9 to 12 due to sufficient self-purification in the river.

Overall, the impact of overflow pollution caused by rainfall in the Liangshui River basin was mainly affected by rainfall, rainfall duration, and rainfall intensity. The shortest time of river quality exceeding the standard was 1 h, and the longest time was for more than 7 days. The overflow pollution generated by continuous rainfall had a great impact on river quality, resulting in a long time of exceedance.

### 3.4. Research on Regulation and Control Scheme

3.4.1. Determination of Control Rainfall

The rainfall corresponding to the CSO control target in Beijing was 33 mm per day, which was heavy rain, according to the rainfall grade, in 24 h (25–50 mm). With only two rainfall events lasting more than 12 h and five rainfall events lasting less than 12 h among the 13 events monitored, rainfall was graded based on the standard of 12 h in the study. For better support of the management of CSO in Beijing, a similar rainfall grade was selected as the control objective. The average rainfall of 23 mm in rainfall event 4 was set as the control for the following reasons: firstly, the rainfall duration was about 1 h, typical of Beijing, with short duration and high intensity; secondly, the range of rainfall in

ten precipitation stations was from 17 to 29 mm, with a standard deviation of only 4.2 and of little difference in the spatial distribution; thirdly, the start and end times of rainfall were basically the same in all precipitation stations.

### 3.4.2. Calculation of Overflow Volume and Load

Values of the overflow flow and water quality of the pre-treatment outlet were monitored in the research. The simple-treatment overflow flow was also monitored; the water quality refers to the pre-treatment overflow. According to the rainfall and equations shown in Figure 5, the CSO coefficient calculated was 0.21 of combined sewer overflow in rainfall of 23 mm; the overflow volume was calculated based on Equation (3) and each catchment area. The overflow load was calculated by the average EMC value of CSO1, CSO2, and CSO3, which was referred to as the water quality in the combined sewer network; the calculations are shown in Table 5.

**Table 5.** Calculation results of the overflow volume and overflow load of each link.

| Overflow Link | No. | Area of Corresponding Upstream Catchment (hm$^2$) | Overflow Volume (m$^3$) | Percentage of Overflow % | Overflow Load (kg) | Percentage of Overflow Load % |
|---|---|---|---|---|---|---|
| Combined pipe network outlet overflow | 21 | 391.6 | 23,849 | 11.3 | 5175 | 8.5 |
| | 2 (CSO1) | 238.0 | 14,494 | 6.9 | 3145 | 5.2 |
| | 22 (CSO3) | 152.6 | 9292 | 4.4 | 2016 | 3.3 |
| | 20 | 148.7 | 9056 | 4.3 | 1965 | 3.2 |
| | 4 | 147.7 | 8997 | 4.3 | 1952 | 3.2 |
| | 10 | 133.6 | 8139 | 3.8 | 1766 | 2.9 |
| | 18 | 120.8 | 7358 | 3.5 | 1597 | 2.6 |
| | 9 | 94.0 | 5724 | 2.7 | 1242 | 2.0 |
| | 8 | 71.0 | 4324 | 2.0 | 938 | 1.5 |
| | 3 | 64.7 | 3940 | 1.9 | 855 | 1.4 |
| | 7 | 57.0 | 3469 | 1.6 | 753 | 1.2 |
| | 23 | 54.4 | 3316 | 1.6 | 720 | 1.2 |
| | 6 | 51.2 | 3117 | 1.5 | 676 | 1.1 |
| | 1 | 46.7 | 2844 | 1.3 | 617 | 1.0 |
| | 5 | 44.8 | 2728 | 1.3 | 592 | 1.0 |
| | 14 | 35.1 | 2135 | 1.0 | 463 | 0.8 |
| | 16 (CSO2) | 34.7 | 2115 | 1.0 | 459 | 0.8 |
| | 19 | 34.5 | 2104 | 1.0 | 457 | 0.7 |
| | 13 | 31.3 | 1903 | 0.9 | 413 | 0.7 |
| | 12 | 17.1 | 1043 | 0.5 | 226 | 0.4 |
| | 11 | 12.3 | 746 | 0.4 | 162 | 0.3 |
| | 17 | 9.8 | 596 | 0.3 | 129 | 0.2 |
| | 15 | 9.1 | 557 | 0.3 | 121 | 0.2 |
| | Subtotal | 2000.7 | 121,849 | 57.6 | 26,441 | 43.4 |
| Pre-treatment overflow | CSO0 | 13,358.3 | 51,935 | 24.6 | 19,943 | 32.8 |
| Simple-treatment overflow | W1 | 18,879.8 | 37,727 | 17.8 | 14,487 | 23.8 |
| Totally | / | / | 211,511 | 100 | 60,871 | 100 |

According to the analysis, the overflows from the combined pipe network, the pre-treatment, and the simple treatment accounted for 57.6%, 24.6%, and 17.8%, respectively, and the pollutant load accounted for 43.4%, 32.8%, and 23.8%, respectively.

### 3.4.3. Construction of the Relationship Curve between the Control Scheme and the Duration of Water Quality Exceeding the Standard

A total of 25 schemes were formed according to the order of overflow load in each outlet, from largest to smallest, which started from the pre-treatment overflow, then gradually increased in regulation and control scales. Scheme 1 regulated CSO0; Scheme 2 regulated CSO0 and W1; Scheme 3 regulated CSO0, W1, and -21; Scheme 4 regulated CSO0, W1, -21,

and -2 (CSO1), and so on, until Scheme 25, with all overflow links regulated. The schemes are summarized in Table 6.

**Table 6.** Summary of regulation scheme.

| Overflow Link | Scenario 1 | | Scenario 2 | | Scenario 3–24 | | Scenario 25 | |
|---|---|---|---|---|---|---|---|---|
| | Regulation Volume (m³) | Regulation Load (kg) | Regulation Volume (m³) | Regulation Load (kg) | Regulation Volume (m³) | Regulation Load (kg) | Regulation Volume (m³) | Regulation Load (kg) |
| Combined pipe network outlet overflow | 0 | 0 | 0 | 0 | 23,849–121,292 | 5175–26,320 | 121,849 | 26,441 |
| Pre-treatment overflow | 51,935 | 19,943 | 51,935 | 19,943 | 51,935 | 19,943 | 51,935 | 19,943 |
| Simple-treatment overflow | 0 | 0 | 37,727 | 14,487 | 37,727 | 14,487 | 37,727 | 14,487 |
| Totally | 51,935 | 19,943 | 89,662 | 34,430 | 113,511–210,954 | 39,605–60,751 | 211,511 | 60,871 |

The hourly pollutant load of the river was calculated by the river flow and water quality; then, the total pollutant load was calculated to be 91,745 kg during exceedance, and the proportions of hourly pollutant load to the total could also be calculated. Hence, pollutant loads of different control schemes were reduced based on these proportions; then, the hourly river quality and exceeded duration were obtained by controlling the regulatory schemes. The relationship curve of the exceeded durations in 25 schemes is shown in Figure 8.

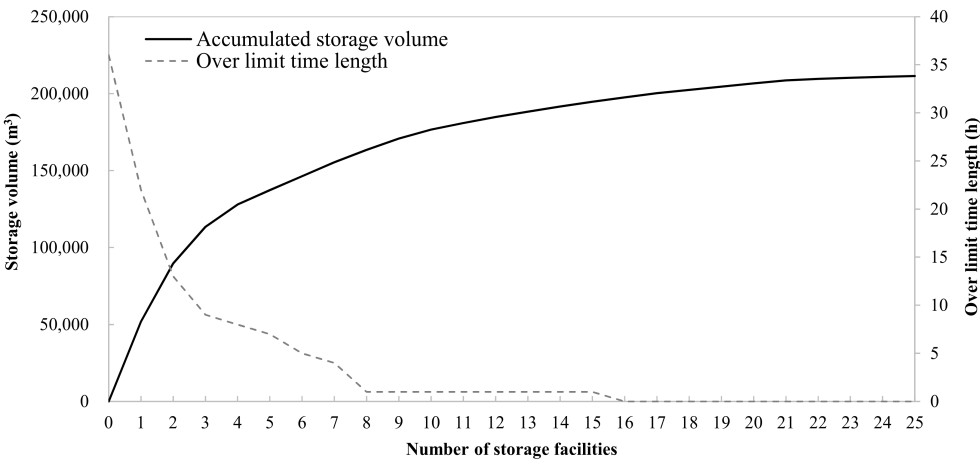

**Figure 8.** River quality improvement effect of different regulation schemes.

3.4.4. Analysis of Regulation and Control Schemes

To reach the standard of river quality under the rainfall of 23 mm, at least 16 storage facilities should be built, with storage of more than 195,000 m³. At least three overflow outlets, CSO0, W1, and -21, should be controlled to ensure exceeded durations of less than 12 h. If the exceeded durations were extended to 24 h, the standard can only be achieved by adopting Scheme 1 for controlling CSO0. The result indicates that the quantity and volume of storage will increase when the control objectives of water quality are improved.

As the impact of CSOs on the environment is location-specific and requires decision-making on their appropriate management at the catchment level [28], in view of river

management, Schemes 3 and 8 were more economical with exceeded durations of 9 and 1 h, respectively. Before Scheme 8, the exceeded duration of river quality was correspondingly shortened with an increase in storage. In contrast, until Scheme 15, the exceeded duration remained unchanged in spite of the storage being increased by 31,000 m3. Therefore, Schemes 9–15 and 17–25 were inapplicable in terms of the input–output ratio. In their specific construction, storage facilities close by in distance could be merged to reduce costs in construction and maintenance. For example, storage facilities numbered 11, 12, 19, and 20 of the combined pipe network could be built together, and the storage scale of CSO0 could be combined with the sewage treatment plant, which could control the CSO0 overflow by regulating the water level of the plant in rainfall. This view is consistent with CSO management, which demands a more systematic strategy for the selection of locations [28].

### 4. Conclusions

Aiming at the control of CSO pollution, a set of comprehensive methods based on the synchronous monitoring of PSR was established (from the aspects of the overflow monitoring scheme, characteristic analysis, a calculation model, and a control target and regulation scheme) and then applied to the Liangshui River basin. The main conclusions are as follows:

(1) Synchronous monitoring was carried out on the combined pipe network, the pre-treatment overflow, the simple-treatment overflow, and the river in the combined sewer system covering the central section of Liangshui River. Thirteen rainfall data sets were obtained, covering light rain, moderate rain, heavy rain, rainstorms, and heavy rainstorms. The rainfall thresholds of the pre-treatment outlet, the combined pipe network, and the simple-treatment outlet were 9, 14, and 16 mm, respectively. The overflow was positively correlated with rainfall, and the correlation decreased with an increase in the catchment areas of outlets. The water quality of the pre-treatment overflow was far higher than the combined system overflow, and COD concentration in heavy rain was higher than in rainstorms.

(2) The shortest time of water quality exceeding the standard, caused by the overflow of pollution in Liangshui River, was 1 h, and the longest time was for more than 7 days. Continuous rainfall had a great impact on river quality, which would exceed the standard for a long time, so it is necessary to carry out CSO control to improve the water quality of the Liangshui River.

(3) Under the condition of controlled rainfall (23 mm rainfall in 12 h), overflow loads of outlets, pre-treatment, and simple treatment accounted for 43.4%, 32.8%, and 23.8%, respectively, and the pollutant load accounted for 66.3% of the total in the period exceeding the standard in the river.

(4) To reach the standard of river quality in rainfall, 16 of the 25 outlets need to be regulated; the total storage scale was only 200,000 m$^3$. If the exceeded duration of river quality is allowed to be extended, the amount of storage required will be greatly reduced. According to the input–output ratio, the regulation scheme of the top three outlets in overflow load was the best, followed by schemes of the top eight outlets, with exceeded durations of 9 and 1 h, respectively. In specific construction, storage facilities close by in distance or with linkage relationships can be merged to save investment and maintenance costs.

**Author Contributions:** Writing, L.Y.; Conceptualization, X.P. and Q.M.; Data analysis, L.Y., S.Y. and M.Y.; Investigation, Y.Y.; Paragraph, J.L. All authors have read and agreed to the published version of the manuscript.

**Funding:** This research was funded by the Major Science and Technology Program for Water Pollution Control and Treatment, Ministry of Ecology and Environment, 2017ZX07103-002, 2017ZX07103-007.

**Institutional Review Board Statement:** Not applicable.

**Informed Consent Statement:** Not applicable.

**Data Availability Statement:** Data are contained within the article.

**Acknowledgments:** Thanks to the Beijing Water Authority and the Beijing Hydrological Station for providing rainfall, river flow, and water quality data.

**Conflicts of Interest:** The authors declare no conflict of interest.

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
