# Peer review of "Research on the Comprehensive Regulation Method of Combined Sewer Overflow Based on Synchronous Monitoring—A Case Study"

_water, doi:10.3390/w14193067_

Round 1

Reviewer 1 Report

 On account of the manuscript WATER-1845041, entitled “Research on the comprehensive regulation method of combined sewer overflow based on synchronous monitoring of the pipe network, sewage plant and river course—A case study of Liangshui River Basin in Beijing” by Lei Yu et al., the authors investigated the set of comprehensive regulation and control methods for river water quality based on the synchronous monitoring of the pipe network, sewage plant, and river course (PSR) in Beijing, China. The topic is important to conduct environmental management of combined sewer overflow (CSO) in the river water environment. After careful consideration, I feel that this manuscript is to be published after improvement of some minor shortcomings. Details of my comments are as follows:

 The manuscript is well written and easy to follow, and the authors got interesting results. Several revisions are, however, required before publication. The present Abstract was not informative. Abstract should include purpose of the research, principal results and major conclusions in a summarized way. In addition, due to separation of the Abstract from the major article, it must be a key to lead readers to evoke a spirit of challenge to contact with the contents of the report. Therefore, the authors are better to improve the Abstract. Another notable aspect is in the ‘1. Introduction’. Although the authors mentioned the aim of this study, the new aspect or view point focused on this research was not clearly stated in the manuscript. Therefore, the authors are strongly encouraged to mention the new points and/or novel aspects or viewpoints which surpass the previous researches in the manuscript clearly. The authors are encouraged to improve these points for enhancement of the novelty and better understanding of the results. After that I am ready to recommend the present manuscript for publication.

Author Response

Thank you for your comments concerning our manuscript (ID: water-1845041), those comments are all valuable and very helpful for revising and improving our paper, and of vital importance to our researches. We have proofread the whole manuscript carefully and revised errors in words and contents, and rewritten the abstract, making the purpose of the research, principal results and major conclusions clear. For a better understanding of the research, we have revised the introduction carefully and enhanced the novel viewpoints of the results, and rearranged unreasonable subsections. Revised portion are marked in blue in the manuscript. The response is as follows:

Point 1: The manuscript is well written and easy to follow, and the authors got interesting results. Several revisions are, however, required before publication.

(1) The present Abstract was not informative. Abstract should include purpose of the research, principal results and major conclusions in a summarized way. In addition, due to separation of the Abstract from the major article, it must be a key to lead readers to evoke a spirit of challenge to contact with the contents of the report. Therefore, the authors are better to improve the Abstract.

(2) Another notable aspect is in the ‘1. Introduction’. Although the authors mentioned the aim of this study, the new aspect or view point focused on this research was not clearly stated in the manuscript. Therefore, the authors are strongly encouraged to mention the new points and/or novel aspects or viewpoints which surpass the previous researches in the manuscript clearly. The authors are encouraged to improve these points for enhancement of the novelty and better understanding of the results. After that I am ready to recommend the present manuscript for publication.

Response 1: We appreciate the reviewer’s suggestion! (1) We have rewritten the abstract carefully, and added the purpose of the research, principal results and major conclusions in a summarized way, which highlights the novelty of the whole study. (2) For a better understanding of the research, we have revised the introduction carefully and enhanced the novel viewpoints of the results.

Reviewer 2 Report

The article Research on the comprehensive regulation method of combined sewer overflow based on synchronous monitoring—A case study.  Liangshui River Basin in Beijing is suitable for publication in WATER. I have several comments to the authors. I am sure, that they should have no problems with improving their article before publication.

 Title

In my opinion your title is too long. My proposition is: Research on the comprehensive regulation method of combined sewer overflow based on synchronous monitoring —A case study.

Abstract

This is very important part of scientific paper-it should be rewritten. 90% of your abstract are conclusions. Please give only chosen results and some information about your research (idea, materials etc.)

Line 14- please avoid such own abbreviations like PSR.

Line 21 Class V standard -this is not clear to use such term in abstract. Just write about in Results chapter.

Introduction

Line 62 bacterial contamination instead bacterial concentration    

Lines 98- law of overflow- this is not clear for me, just explain it please.

Line 105-monitoring of power plants? What power plants?

Maybe it is a good idea to present clearer scientific and practical aim of your research and article in the end of chapter introduction?

Material and methods

I suggest to change of sub chapters- first give Overview of the study area and next Technical process and calculation methods……

Figure 1 is important but it is to complicated. This is not enough to write only: The technical flow chart is shown 129 in Figure 1. The following parts of the figure should be described or figure should be simplified

Please use term term Equation rather than Formula

Line 188-190

“water plants” or it should be wastewater treatment plants- please explain it

Line 209

Please do not use such a term as garbage- what garbage?

Table 1- In my opinion data given in this table are important.

Table 2- improve table editing. Why time is not the same in all of monitoring data?

Results and discussion

Table 3 Please remove column “Rainfall grade (in 12h)  This is not important

Line 289-290- bring it to the conclusions chapter please.

My main comment on the whole text is as follows: In the Results and Discussion section, there is only a description of the research results. What is missing is a discussion, that is, a reference of one's own results to the literature related to the topic. I think this chapter should be improved before publishing in Water. I believe there will be no problem to do it.

Author Response

Thank you for your comments concerning our manuscript (ID: water-1845041), those comments are all valuable and very helpful for revising and improving our paper, and of vital importance to our researches. We have proofread the whole manuscript carefully and revised errors in words and contents, and rewritten the abstract, making the purpose of the research, principal results and major conclusions clear. For a better understanding of the research, we have revised the introduction carefully and enhanced the novel viewpoints of the results, and rearranged unreasonable subsections. Revised portion are marked in blue in the manuscript. The response is as follows:

Point 1: Title—In my opinion your title is too long. My proposition is: Research on the comprehensive regulation method of combined sewer overflow based on synchronous monitoring —A case study.”

Response 1: We appreciate the reviewer’s suggestion! We have revised the title in the manuscript as suggested.

Point 2: Abstract—(1) This is very important part of scientific paper-it should be rewritten. 90% of your abstract are conclusions. Please give only chosen results and some information about your research (idea, materials etc.). (2) Line 14- please avoid such own abbreviations like PSR. (3) Line 21 Class V standard -this is not clear to use such term in abstract. Just write about in Results chapter.

Response 2: We appreciate the reviewer’s suggestion! (1) We have rewritten the abstract carefully in accordance with the requirement of the scientific paper, simplified conclusions and added the research (idea, materials etc.) in this part. (2) We have revised the Line 14 in the manuscript as suggested. (3) To make it clear, we have changed the term of Class V standard to national standard in abstract, and just write about Class V in Results chapter.

Point 3: Introduction—(1) Line 62 bacterial contamination instead bacterial concentration. (2) Lines 98- law of overflow- this is not clear for me, just explain it please. (3) Line 105-monitoring of power plants? What power plants? (4) Maybe it is a good idea to present clearer scientific and practical aim of your research and article in the end of chapter introduction?

Response 3: We appreciate the reviewer’s suggestion! (1) We have revised the Line 62 as suggested. (2) The Lines 98- law of overflow, namely the regularity of overflow, and we have changed this word in the manuscript to make the expression clear. (3) Line 105-monitoring of power plants, namely sewage plants, and we have revised this word in the sentence. (4) We have revised the part of introduction, which presented clearer scientific and practical aim of the research and article in the end of the chapter introduction.

Point 4: Material and methods—(1) I suggest to change of sub chapters- first give Overview of the study area and next Technical process and calculation methods……(2) Figure 1 is important but it is to complicated. This is not enough to write only: The technical flow chart is shown 129 in Figure 1. The following parts of the figure should be described or figure should be simplified. (3) Please use term term Equation rather than Formula. (4) Line 188-190-“water plants” or it should be wastewater treatment plants- please explain it. (5) Line 209-Please do not use such a term as garbage- what garbage? (6) Table 1- In my opinion data given in this table are important. (7) Table 2- improve table editing. Why time is not the same in all of monitoring data?

Response 4: We appreciate the reviewer’s suggestion! (1) We have changed of sub chapters in the manuscript, which first given the Overview of the study area, and next Technical process and calculation methods. (2) To make the expression of technical flow more clear, we have simplified the Figure of Technical process of CSO comprehensive regulation method based on synchronous monitoring of PSR. (3) We have changed the Formula to Equation as suggested. (4) Water plants are four wastewater reclamation plants which signed in figure of Location and Monitoring Points of Liangshui River Basin. (5) This garbage is one type of the domestic waste, and we have revised this term in the manuscript. (6) We have added both the area and proportion of each type of underlying surface in the Table of Information of drainage zoning (water collection range) corresponding to each monitoring point. (7) We have edited the Table of Summary of monitoring data, and revised the time of monitoring data. The data in the Table are different types of monitoring data, including rainfall Data, flow monitoring data, water quality, simply-treatment overflow volume and river water quality and flow, which monitored by different equipment and time series and monitored during 2020.7-2020.9.

Point 5: Results and discussion—(1) Table 3 Please remove column “Rainfall grade (in 12h)  This is not important. (2) Line 289-290- bring it to the conclusions chapter please. (3) My main comment on the whole text is as follows: In the Results and Discussion section, there is only a description of the research results. What is missing is a discussion, that is, a reference of one's own results to the literature related to the topic. I think this chapter should be improved before publishing in Water. I believe there will be no problem to do it.

Response 5: We appreciate the reviewer’s suggestion! (1) We have removed column “Rainfall grade (in 12h) in Table of Summary of rainfall characteristics and overflow situation. (2) We have added the Line 289-290 in the conclusions chapter. (3) In the Results and Discussion section, we have added the reference cited to the literature related to the topic, and improved the whole manuscript as suggested.

Reviewer 3 Report

The paper presents the results of an extensive data collection campaign, aimed at evaluating the effects of the combined sewer overflow on river water quality.

The paper is very poorly written, it is required extensive proofreading in my opinion. Besides the English language, the style of the paper leaves much to be desired:

- the abstract appears delirious, with a list of results referring to specific elements of the network that are introduced later in the paper. Remember that the abstract is supposed to be the first part to be read, it should give a general idea of the aims, methods and findings, to convince the reader that it is worth reading the paper, rather than confusing him.

- some subsections should be completely rearranged, in my opinion. As an example, subsection 2.1.1 seems to be structured more like a tutorial than like a report, with a sequence of actions (Carry out data collection.., Determine the control rainfall.., calculate the river water quality..)

- Please use a proper measurement unit for frequency. What does 10 min/time mean? Did you mean 0.1 min^-1?

- In line 318, what do you mean by "the accuracy of the model meets the research needs"? Did you define a threshold? Please define it or add references, otherwise, it is better to remove that statement.

Author Response

Thank you for your comments concerning our manuscript (ID: water-1845041), those comments are all valuable and very helpful for revising and improving our paper, and of vital importance to our researches. We have proofread the whole manuscript carefully and revised errors in words and contents, and rewritten the abstract, making the purpose of the research, principal results and major conclusions clear. For a better understanding of the research, we have revised the introduction carefully and enhanced the novel viewpoints of the results, and rearranged unreasonable subsections. Revised portion are marked in blue in the manuscript. The response is as follows:

Point 1: The paper is very poorly written, it is required extensive proofreading in my opinion.

Response 1: We appreciate the reviewer’s suggestion! We have proofread the whole manuscript carefully and revised errors in words and contents.

Besides the English language, the style of the paper leaves much to be desired:

Point 2: The abstract appears delirious, with a list of results referring to specific elements of the network that are introduced later in the paper. Remember that the abstract is supposed to be the first part to be read, it should give a general idea of the aims, methods and findings, to convince the reader that it is worth reading the paper, rather than confusing him.

Response 2: We appreciate the reviewer’s suggestion! We have rewritten the abstract carefully, and added the general idea of the aims, methods and findings in the research.

Point 3: Some subsections should be completely rearranged, in my opinion. As an example, subsection 2.1.1 seems to be structured more like a tutorial than like a report, with a sequence of actions (Carry out data collection.., Determine the control rainfall.., calculate the river water quality..)

Response 3: We appreciate the reviewer’s suggestion! We have revised the whole manuscript carefully and rearranged unreasonable subsections, including the subsection of technical process, which meets the requirement of the report.

Point 4: Please use a proper measurement unit for frequency. What does 10 min/time mean? Did you mean 0.1 min^-1?

Response 4: We appreciate the reviewer’s suggestion! The 10 min/time means 10 min^-1, and we have revised the measurement unit for frequency carefully in the whole manuscript.

Point 5: In line 318, what do you mean by "the accuracy of the model meets the research needs"? Did you define a threshold? Please define it or add references, otherwise, it is better to remove that statement.

Response 5: We appreciate the reviewer’s suggestion! We have not defined the threshold, and this statement have been removed.

Round 2

Reviewer 3 Report

I'm glad to see the abstract and section 2.1.1 have been improved.

However, the draft still requires extensive proofreading, possibly from a native English speaker, because there are some really weird passages. Here are a couple of examples:

- "event means concentration" is a sentence, you are basically writing that the meaning of the word event is concentration. You should write "event mean concentration" instead, coherently with the existing literature.

-  "the duration of five heavy drops of rain was less than 12 hours" doesn't make much sense. You should rather call them rainfall events, a heavy drop of rain is just a big single drop!

It is clear that you did not mean what you have written in those examples, and in other passages of the manuscript, and that must be fixed.

I've noticed that you have changed the measurement units for frequency without converting the numbers. Now the values in subsection 2.3 don't seem to match table 2 if I understand what you are writing well. You write that with the quantum dot spectral sensing technology you have been reading 10 water quality values per minute, i.e. a resolution of 6 seconds, but in the table, the resolution range is coarser (5 min - 0.5h). Which one is correct?
Same issue with serial number 5: frequency of 1-2 per hour in the text, resolution of 1-2 hour in the table. Please remember that frequency is measured in counts per unit of time whilst the measurement unit of time resolution is the time itself! A sampling frequency of 10 min^-1 generates a resolution of 0,1 minutes.

Author Response

Thank you for your comments concerning our manuscript (ID: water-1845041) and sorry for the late response, those comments are all valuable and very helpful for revising and improving our paper. During this period, we have an extensive proofreading in the whole manuscript carefully and revised weird sentences, then reedited all chapters. Revised portions are marked in blue in the manuscript. The response is as follows:

Point 1: I'm glad to see the abstract and section 2.1.1 have been improved. However, the draft still requires extensive proofreading, possibly from a native English speaker, because there are some really weird passages. Here are a couple of examples:

- "event means concentration" is a sentence, you are basically writing that the meaning of the word event is concentration. You should write "event mean concentration" instead, coherently with the existing literature.

Response 1: We appreciate the reviewer’s suggestion! We have revised this error in the whole manuscript.

-  "the duration of five heavy drops of rain was less than 12 hours" doesn't make much sense. You should rather call them rainfall events, a heavy drop of rain is just a big single drop!

Response 2: We appreciate the reviewer’s suggestion! We have revised this error of rainfall events in the paper.

It is clear that you did not mean what you have written in those examples, and in other passages of the manuscript, and that must be fixed.

Response 3: We appreciate the reviewer’s suggestion! We have rewritten passages of the manuscript carefully.

I've noticed that you have changed the measurement units for frequency without converting the numbers. Now the values in subsection 2.3 don't seem to match table 2 if I understand what you are writing well. You write that with the quantum dot spectral sensing technology you have been reading 10 water quality values per minute, i.e. a resolution of 6 seconds, but in the table, the resolution range is coarser (5 min - 0.5h). Which one is correct? Same issue with serial number 5: frequency of 1-2 per hour in the text, resolution of 1-2 hour in the table. Please remember that frequency is measured in counts per unit of time whilst the measurement unit of time resolution is the time itself! A sampling frequency of 10 min^-1 generates a resolution of 0,1 minutes.

Response 4: We appreciate the reviewer’s suggestion! We have revised the expression of subsection 2.3 and table 2. In the monitoring process, the weir-type flowmeter was selected to monitor the flow of outlets with every 5 minutes, and a self-developed intelligent sampler (patent No.: ZL201720745546.9) was selected to monitor the water quality. When the overflow occurs from the outlet, the automatic sampler collects the effluent stored in a 500 mL polyethylene bottle. The sample collection interval was set: Samples shall be taken every 5 minutes in the first half an hour after overflow, and every 10 minutes from half an hour to 1 hour, then every 30 minutes after 1 hour of overflow, until all the 24 bottles in the sampler were collected or the overflow was over.

Round 3

Reviewer 3 Report

The text is now adequately readable in my opinion.

Just pay attention to the difference between frequency and time resolution: when you refer to a time unit (seconds, minutes etc...) you should call it a resolution, whilst the frequency has a time^-1 unit (Hertz, counts per hour etc...).

Then, in the last column of table 2 you should convert them in time^-1 or call it resolution.

Author Response

Thank you for your comments concerning our manuscript (ID: water-1845041) during all this time, those comments are of vital importance to improve our paper. Revised portions are marked in blue in the manuscript. The response is as follows:

Point 1: The text is now adequately readable in my opinion.

Just pay attention to the difference between frequency and time resolution: when you refer to a time unit (seconds, minutes etc...) you should call it a resolution, whilst the frequency has a time^-1 unit (Hertz, counts per hour etc...).

Then, in the last column of table 2 you should convert them in time^-1 or call it resolution.

Response 1: We appreciate the reviewer’s suggestion! We have revised the last column of table 2 called it "Resolution" in the manuscript, and we will pay more attention to the difference between frequency and time resolution in our papers.
